# Development of a Human–Robot Interface for Cobot Trajectory Planning Using Mixed Reality

Raúl Calderón-Sesmero [1,*], Jaime Duque-Domingo [2], Jaime Gómez-García-Bermejo [1,2] and Eduardo Zalama [1,2]

1    CARTIF, Technological Center, 47151 Valladolid, Spain; jaigom@uva.es (J.G.-G.-B.); ezalama@uva.es (E.Z.)
2    ITAP-DISA, University of Valladolid, 47011 Valladolid, Spain; jaime.duque@uva.es
*    Correspondence: raucal@cartif.es

**Abstract:** The growing demand for projects with collaborative robots, known as "cobots", underlines the need to efficiently address the execution of tasks with speed and flexibility, without neglecting safety in human–robot interaction. In general terms, this practice requires knowledge of robotics programming and skill in the use of hardware. The proposed solution consists of a mixed reality (MR) application integrated into a mixed reality head-mounted device (HMD) that accelerates the process of programming the complex manoeuvres of a cobot. This advancement is achieved through voice and gesture recognition, in addition to the use of digital panels. This allows any user, regardless of his or her robotics experience, to work more efficiently. The Robot Operating System (ROS) platform monitors the cobot and manages the transfer of data between the two. The system uses QR (Quick Response) codes to establish a precise frame of reference. This solution has proven its applicability in industrial processes, by automating manoeuvres and receiving positive feedback from users who have evaluated its performance. This solution promises to revolutionize the programming and operation of cobots, and pave the way for efficient and accessible collaborative robotics.

**Keywords:** mixed reality (MR); collaborative robots; voice and gesture recognition; trajectory planning; human–robot interaction (HRI)

## 1. Introduction

The substantial growth of collaborative robotics in recent decades is attributable to its remarkable features. Collaborative robots, or "cobots", significantly outperform industrial robots in terms of flexibility and safety, enabling human–robot interaction (HRI) [1]. It is undeniable that the hand-guided programming of cobots outperforms the programming process through coding in several respects [2]. On the one hand, this work is often tedious due to the inertia and uncontrolled movements of the cobot's joints [3]. Furthermore, this practice is nuanced as hand-guidance is influenced by the dimensions and weight of the cobots [4]. On the other hand, operators must have specific knowledge of how to operate the teach pendant. Consequently, many companies invest considerable time and financial resources in training their employees [5].

Industry 4.0 has brought new approaches to cobot programming. These approaches use innovative technologies to improve efficiency and productivity in industry [6]. Some of these methods involve the implementation of gesture control systems for performing basic tasks with collaborative robots [7]. Other methods focus on optimizing collaborative robotic manipulation by predictive control through a remote controller [8]. These approaches provide solutions for controlling cobots remotely. However, they do not solve the challenge of operating them in the same physical workspace. In addition, they may lack intuitiveness in their use.

Teleoperation of collaborative robots in simulated environments effectively solves these problems. This is achieved through a human–robot interface that enables the programming of robotic arms using a game controller. This interface provides the user with

constant feedback on the robot's status, facilitating the performance of various tasks in an easy and intuitive manner [9]. However, the user's perception of the environment has limitations. The user does not have an accurate perception of the robot's range, and the arrangement of elements in the simulation may not match their actual location.

The ability to interpret the environment is significantly enriched by the application of extended reality (XR). This term covers the spectrum of immersive technologies: augmented reality (AR), virtual reality (VR) and mixed reality (MR) [10]. This aspect is crucial here, as it fully integrates the physical and digital environments. This creates a unique environment where digital objects can interact coherently with the user and the physical environment around them [11,12]. MR differs from AR by adding virtual elements to physical reality and manipulating the user's physical sensations directly [13].

Moreover, visualization and interaction with digital cobot twins is made possible by implementing mixed reality [14]. Head-mounted display devices (HMD) using this technology enable the integration of effective and easy-to-understand human–machine interfaces (HMI) [15]. In this way, an operator can perform tasks in a shared environment in which virtual and real robots operate in synchrony to carry out specific tasks [16].

In this scenario, the challenge is to incorporate this technology into industry, and to look for methods of programming that improve both workflow and productivity [17]. In this context, Programming by Demonstration (PbD) emerges as a viable solution. This technique involves the human operator demonstrating the task and then the robotic system executing it [18,19]. The inclusion of human natural language can simplify the PbD process with a robotic arm. This ensures that the system is intuitive and encourages faster task scheduling [20].

This paper presents an innovative solution that simplifies the PbD technique and optimizes path planning in cobots using MR. The system is mainly composed of an HMD as a device to guide and recognize the user during the activity, together with a collaborative robot. By introducing digital elements that enhance the perception of the cobot's reach and guide the user intuitively and safely, HRI is facilitated in a shared workspace. The goal is to adapt this solution to the industrial environment. Irrespective of the cobot's dimensions, this system establishes a common programming standard through MR.

The paper begins with an overview of XR systems for HRI (Section 2), followed by a novel approach involving the use of MR for trajectory planning of a cobot (Section 3). It then discusses the implementation of the proposed system, evaluating safety, user experience, task complexity with the robotic arm, trajectory planning accuracy and time (Section 4). It concludes with the results (Section 5).

## 2. Objectives and Related Work

Previous studies explored employing XR to improve the efficiency of cobots in industrial environments. It is crucial to integrate strategies to establish an optimal system that performs key functions during industrial activity, including:

- Ensuring operator safety;
- Facilitating smooth and efficient communication between the devices involved;
- Guiding the user through an intuitive human–robot interface;
- Enabling path planning in cobots using the user's natural language;
- Achieving a high degree of complexity in the tasks to be performed; and
- Ensuring that the results obtained maintain high standards of accuracy and reduce the programming time of the manoeuvres.

In recent years, several related works have been developed in this context. Some of the studies that focus on the control of cobots by means of human–robot interfaces with HMDs are discussed below.

Rudorfer et al. [21] present a system that focuses on image capture, object recognition and manipulation in an operational environment. The cycle begins with image acquisition of the cobot's workspace, followed by object recognition through the Microsoft HoloLens mixed reality HMD. A programming cycle allows the user to control the robotic arm to

move Lego blocks on a platform bounded by markers. This system allows the user to control the robot efficiently through gestures, making it easier to select and manipulate objects in the cobot's environment. Although the prototype can move objects intuitively, it has limitations in terms of accuracy. Target position determination is based on the intersection of the user's gaze with the platform. This is affected by the initial calibration, object detection and position tracking of the HoloLens, which creates limitations in the process.

Similarly, Blankemeyer et al. [20] devised a programming method that uses AR to simulate the sequence of picking and placing components. This makes it possible for the cobot to reproduce the task. They used the Microsoft HoloLens 2 (HL2) device in the trials. The application employed image recognition using markers, overlaying digital copies of the parts, and continuously tracking their location. The results evidenced significant inaccuracy in task performance that is largely affected by their angle and distance from the HMD.

Araiza-Illan et al. [22] presented a demonstration using HoloLens to intuitively reprogram a robotic packaging application. An AR interface is introduced that enables fast and effective reconfiguration of robots, adapting them to changes in production orders. It uses simple hand gestures and spatial mapping to recognize objects and QR (Quick Response) codes. This facilitates precise location of landmarks and accurate placement of objects on trays. The demonstration illustrates how the packaging application can be reconfigured without requiring prior knowledge of robot programming. This suggests the possibility of implementing further enhancements to offer more parameterization options.

The study by Abir Gallala et al. [1] also demonstrates accuracy in employing Microsoft HoloLens to simulate robots in real environments using MR. The system employs QR codes to visualize accurate models, allowing interaction, simulation and export for implementation on real robots. The system consists of a collaborative robot, a HoloLens as a device to visualize the human–robot interface and a Robot Operating System (ROS) to coordinate the processes. The solution provides accurate simulations, but focuses on manual manipulation of the cobot's joints, which does not speed up programming.

Dimitris Mourtzis et al. [16] used the concept of the aforementioned study. The text details an innovative system for controlling a robotic arm using its digital twin. It consists of software modules that allow three-dimensional (3D) representation of the robot's environment and its functional simulation, which facilitates remote control of the arm. Through a digital interface, safety zones are established. During testing, a significant reduction in errors and assembly time was noted in collaborative environments. In addition, a considerable improvement in user awareness was observed. The need to improve communication between modules and strengthen network security for future deployments is highlighted.

Inês Soares et al. [23] proposed the development of a system that guides non-robotics operators to program a cobot remotely. It uses HL2 to project panels and detect gestures. It includes a safety zone that delimits the robotic arm's workspace. The specified coordinates in the application do not globally match those of the cobot because the digital workspace is not located at the base of the robot. Communication between the application and the cobot is through the ROS. Experiments with geometric shapes validated the robotic arm's ability to reproduce programmed paths. This allows an operator to program a cobot by demonstration using AR. The solution is suitable for applications that tolerate errors down to 1 cm. However, the system only allows path planning by recording coordinates with an entered sampling rate. Consequently, the programmed manoeuvres are highly sensitive to human factors such as body vibration. This prevents the system from being used for industrial manoeuvres.

Some recent studies have conducted comparisons in path planning between handguided and a MR interface integrated into HL2. Chang Wesley et al. [24] and Graziano Carriero et al. [25] conducted tests contrasting these two methods, and concluded that the mixed reality interface is more efficient and intuitive than the hand-guided method. Both studies employ a virtual representation of the robot that is placed in the same position and orientation as the real robot using marker detection. Regarding the path-planning method,

the first abovementioned paper (Chang Wesley et al.) propose setting the waypoints using gaze. This approach is innovative as it significantly reduces physical effort and task completion time by about 25% in comparison to a joystick interface. However, some users have commented that this method is occasionally inaccurate; however, no specific figures have been provided in this regard.

The second abovementioned paper (Graziano Carriero et al.) focuses on the calculation of trajectories from the target position of the Tool Center Point (TCP) of the robotic arm. The waypoints are calculated by an algorithm executed by an ROS node. The main limitation of this method lies in the fact that the algorithm only computes linear paths from the source location to the TCP destination location. However, the mixed reality interface allows for the modification of the coordinates of the waypoints. This paper notes that the system has an error of 5.289 mm with a deviation of 2.76 mm in the calibration phase; this is possibly due to the exclusive use of a QR code in the detection.

In relation to the proposed objectives, the studies analysed offer interesting contributions in certain aspects and present limitations in others. Table 1 summarizes the most significant particularities of the studies analysed with respect to the objectives.

**Table 1.** Comparison of related work according to the key aspects of MR for the industrial environment.

| Ref. | Security | Communication | MR Interface | Planning Method | Range of Manoeuvres | Precision, Accuracy and Time |
|---|---|---|---|---|---|---|
| [21] | Delimited work area | RESTful web service | Superposition of digital 3D models | Gesture-based capture, recognition and manipulation of virtual objects | Collection and placement of Lego blocks | Does not provide exact figures Limited by calibration, detection and HoloLens tracking |
| [20] | - | RSI (based on UDP) | Superposition of digital 3D models Navigation by buttons and voice commands | Gesture-based capture, recognition and manipulation of virtual objects | Assembly of specific components | Best results: Accuracy of 1–2 mm and precision of 3–5 mm. Worst results: Accuracy of 9 mm and precision of 45 mm |
| [22] | - | - | Digital panels with buttons | Definition of locations and objects by means of gestures | Packaging | - |
| [1] | - | MQTT ROS | Superposition of digital 3D models | Cobot digital twin axis control | Point trajectories | - |
| [16] | Visualization of cobot range | ROS WebSocket XML-RPC RTDE | Robot control by sliders | Gestures | Assembly and transfer of objects | 24% reduction in assembly time |
| [23] | Visualization of cobot range | ROS WebSocket | Digital panels with buttons and visualization of the path traced in the planning process | Path registration through gestures in a different workspace of the cobot | Position tracking of the user's index finger without regard to orientation | Accuracy > 1 cm |
| [24] | Check the validity of the calculated trajectory | ROS WebSocket | Visualization of virtual robot Visualization of waypoints by means of spheres Navigation by buttons and voice commands Auditory feedback of robot status | Path points planned by the user's gaze and voice | Point trajectories | Approximately 25% reduction in task execution compared to a joystick interface |
| [25] | Check the validity of the calculated trajectory Visualization of cobot range | ROS WebSocket | Visualization of virtual robot Visualization of waypoints by means of spheres Navigation by buttons | Set the final position of the TCP | Path from origin to set point | In calibration: Accuracy > 5.289 mm Precision > 2.76 mm |

This paper discusses the different aspects of the system describing the methods adopted to meet the objectives described. The approach used is based on capturing the location of the operator's index finger, integrating a digital interface in an HMD that guides and recognizes it during the process. By employing voice and gesture recognition to capture the coordinates of the trajectories, greater efficiency is achieved. These coordinates are then transformed into instructions for the cobot, speeding up workflow and increasing productivity.

## 3. Analysis of the System

The proposed system is based on the creation of a human–robot interface for trajectory planning in a cobot, following the objectives presented above. The system focuses on two main modules. One assists the user in trajectory planning, while the other one manages the communication between the coordinate capture module and the cobot.

The operation of the system is shown in Figure 1. Fundamentally, the structure consists of a closed loop that analyses and transmits the specified coordinates to the collaborative robot. Once the cobot completes a manoeuvre, the user is offered the possibility to repeat the current program in a loop or to plot the points of a new trajectory. If the user indicates a coordinate outside the range of the robotic arm during planning, its transmission is cancelled. At this point, the option is offered to delete the path planned so far or to correct only the last coordinate. The system is terminated as soon as the user closes the application.

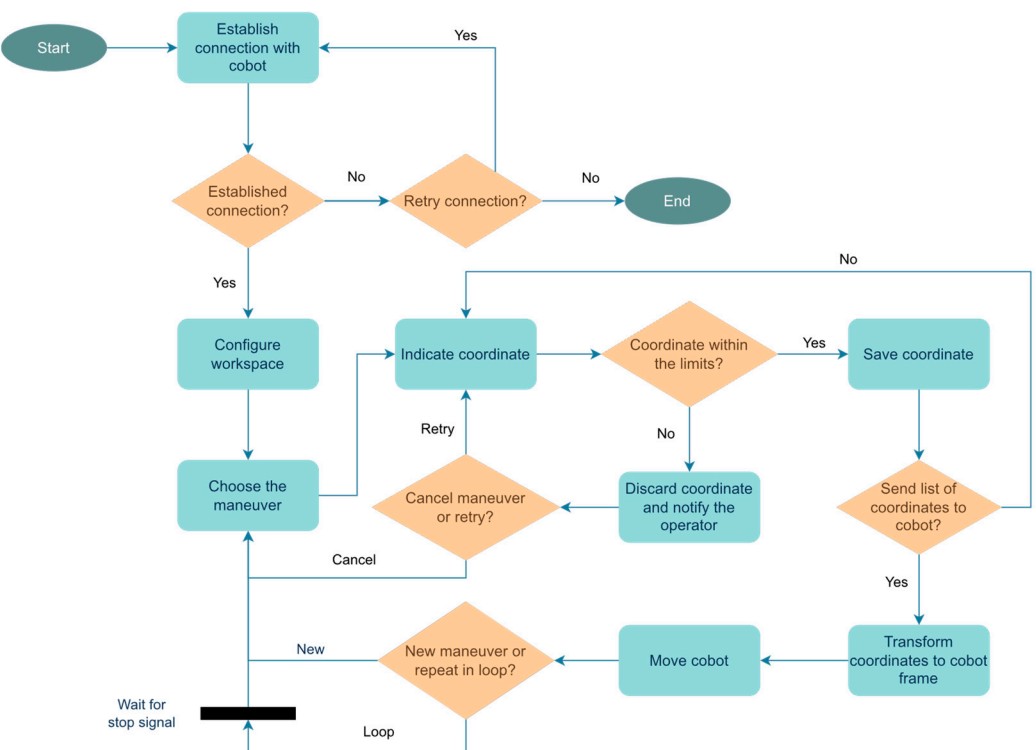

**Figure 1.** Flow diagram of the general operation of the proposed system.

### 3.1. Communication

The system uses a messaging broker as the core of the communication architecture. Using the producer/consumer communication model, software programs hosted on the devices exchange messages on topics. These programs are configured as nodes in the system, and assume the specific roles of publisher and/or subscriber [26].

During the operation of the system, the nodes are responsible for executing the established functions for interaction, transmission and processing of data [27]. This dynamic is illustrated in the schematic depicted in Figure 2, which shows the data flow within the system. One of the nodes subscribes to the topic where the coordinates are published by the

HMD. In this way, it is in charge of translating the user's instructions into programs that are understandable to the cobot. Meanwhile, another node analyses the state of the robotic arm and publishes messages in a topic that are used by the HMD to project digital elements.

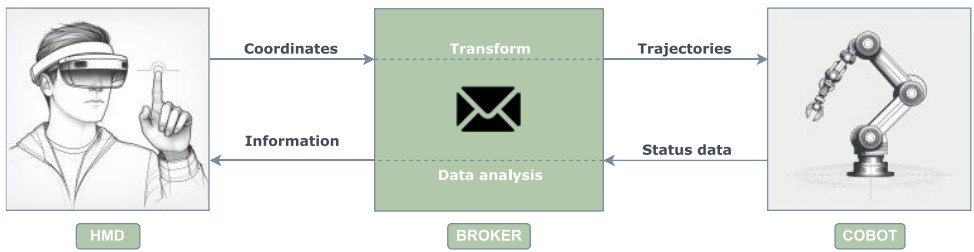

**Figure 2.** The system communication model. The arrows indicate the source and the destination of the data.

### 3.2. HMD—Cobot Calibration

The trajectory planning procedure requires the establishment of a reference frame that enables the conversion of the coordinates acquired in the MR application to the cobot reference. In this way, it is intended to match the workspaces of both references. The method used to achieve this is based on estimating the homogeneous transformation matrix (HTM) between the HMD coordinate system and the cobot. This technique is known as hand–eye calibration [28].

The system requires iterative computation of the matrix at each instance of application initialization. This requirement arises due to the indeterminate nature of the HMD coordinate system since it is based solely on its initial position at application start-up [29]. The resolution adopted is based on linking the head coordinate system (HCS) with the world coordinate system (WCS). The latter is known by the cobot coordinate system (CCS).

In this approach, hand–eye calibration of the systems involves the selection of coincident points in the WCS systems with respect to HCS and CCS. In this way, the HTM that locates the HMD with respect to the cobot frame is obtained ($^{C}\mathbf{T_H}$). Since the HCS is set at application start-up, it is crucial to have enough points for fast and accurate calibration. It has been found that three points are sufficient for accurate calibration and to correctly construct this matrix. The calculation of the matrix $^{C}\mathbf{T_H}$ is described below:

Given three points with respect to HCS: $\mathbf{p1} = (x1, y1, z1)$, $\mathbf{p2} = (x2, y2, z2)$ and $\mathbf{p3} = (x3, y3, z3)$, the unit vectors $\mathbf{u}, \mathbf{v}$ y $\mathbf{w}$, which represent the direction of the X, Y and Z axes of the CCS with respect to the WCS.

$$
\begin{aligned}
\mathbf{u} &= \frac{\mathbf{p1-p2}}{|\mathbf{p1-p2}|} = (ux, uy, uz) \\
\mathbf{p13} &= \frac{\mathbf{p3-p1}}{|\mathbf{p3-p1}|} \\
\mathbf{w} &= \mathbf{u} \times \mathbf{p13} = (wx, wy, wz) \\
\mathbf{v} &= \mathbf{w} \times \mathbf{u} = (vx, vy, vz)
\end{aligned}
\tag{1}
$$

Once the HCS axes are calculated, a coordinate origin (**o**) is determined. In this case, **o** is equal to the point **p1**:

$$
\mathbf{o} = \mathbf{p1} = (ox, oy, oz)
\tag{2}
$$

Therefore, the HTM that locates the HCS with respect to the WCS reference ($^{W}\mathbf{T_H}$) is obtained as follows:

$$
^{W}\mathbf{T_H} = \begin{pmatrix} ux & vx & wx & ox \\ uy & vy & wy & oy \\ uz & vz & wz & oz \\ 0 & 0 & 0 & 1 \end{pmatrix}
\tag{3}
$$

This method is applied in the same way to obtain the HTM representing the CCS with respect to the WCS ($^{W}T_{C}$); with the same three points known in CCS. Finally, by the ratio of both homogeneous transformation matrices $^{C}T_{H}$ is calculated:

$$^{C}T_{H} = {}^{C}T_{W} \cdot {}^{W}T_{H} \tag{4}$$

where $^{C}T_{W}$ is equal to the inverse matrix of $^{W}T_{C}$:

$$^{C}T_{W} = {}^{W}T_{C}{}^{-1} \tag{5}$$

Therefore, a point captured by the HMD $^{H}\mathbf{p} = (hx, hy, hz, 1)$ is transformed to a point $^{C}\mathbf{p} = (cx, cy, cz, 1)$ in CCS, as follows:

$$^{C}\mathbf{p} = {}^{C}T_{H} \cdot {}^{H}\mathbf{p} \tag{6}$$

This process is performed to combine the HCS and CCS spaces, with the objective of directing the cobot to the coordinates specified by the user.

### 3.3. MR Interface

The user follows the indications presented in the application integrated in the HMD by means of gestures and voice commands. A series of digital elements is displayed to guide the user to carry out the programming process correctly.

During trajectory planning, when entering a coordinate, it is verified that the cobot can reach it properly. Therefore, a check is performed to ensure that the specified coordinate is within the workspace of the collaborative robot. This procedure consists of two steps:

i.    The verification that a point lies within the cobot's workspace. For example, if we define the cobot workspace as a cylinder of diameter (dc) and height (hc), inserted in the centre of a sphere of diameter (ds), larger than dc, a point $^{H}\mathbf{p} = (px, py, pz)$ is considered within this zone if it fulfils certain conditions.

$$\begin{cases} \sqrt{(px - ox)^2 + (py - oy)^2 + (pz - oz)^2} \leq \frac{ds}{2} \\ \sqrt{(px - ox)^2 + (pz - oz)^2} \geq \frac{dc}{2} \; ; \; \text{if } |py| < \frac{hc}{2} \\ \sqrt{(px - ox)^2 + (pz - oz)^2} \leq \frac{dc}{2} \; ; \; \text{if } |py| \geq \frac{hc}{2} \end{cases} \tag{7}$$

where ox, oy and oz represent the 3D coordinates of the centre of the cobot base in the HCS. Figure 3 shows the representation of the cobot workspace described with the example coordinates and the reference system used for the calculation of the equations.

ii.   The verification that the movement of the chain of joints of the robotic arm to move the TCP to the specific position has a solution. In other words, it is to verify the feasibility of the inverse kinematics of the cobot [30]. This verification ensures that the cobot can reach the position specified from its initial joint configuration. There are several methods to calculate the inverse kinematics of a robotic arm. However, the use of programming libraries and software greatly simplifies these calculations, and this becomes a fundamental tool to obtain the desired solution.

If both conditions are met, the item is checked and published in the corresponding topic for further use.

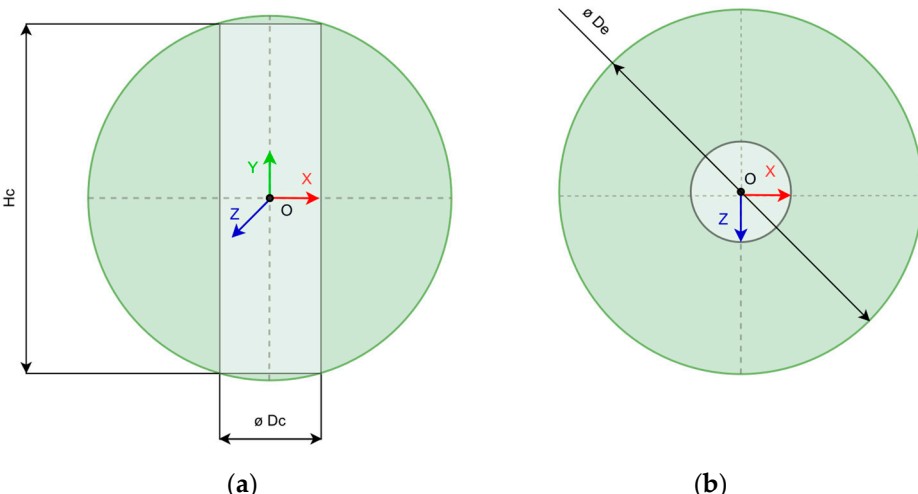

**Figure 3.** Representation of the workspace of a reference cobot. (**a**) Front view; (**b**) Aerial view.

*3.4. Planning and Sending of Trajectories*

The trajectory planning process is based on the identification of the index finger of the user's right hand using the HMD to find its coordinates in the HCS. To enhance the user's perception with the system, a digital element is projected, overlaid on the finger. This acts as a reference point for the location, which is tracked by the application.

The user utters voice commands to post a message with the programmed coordinates in the corresponding topic of the messaging broker. Once the subscriber node receives the list of coordinates, it proceeds to transform them to the cobot reference using the Formula (6). Subsequently, it prepares the program that is sent to the cobot.

The combination of hand detection and the user's voice enables path planning using human natural language. This provides an intuitive and easy way to create trajectories.

**4. Experiments and Results**

The human–robot interface has been developed on the Unity platform, which allows the use of the Mixed Reality Toolkit (MRTK) framework. This interface is hosted on the HL2 device. The HL2 is an advanced device equipped with a second-generation holographic processing unit and a perception system utilizing cameras for localization, eye tracking and gesture recognition. This sophisticated design enables the device to harness contemporary algorithms for voice and gesture recognition, providing users with an advanced and versatile augmented reality experience [31,32]. Additionally, the HL2 boasts a reclining clear lens system, rendering it well-suited for deployment in industrial environments.

The cobot used is the UR5e from Universal Robots. The transmission of information between the HL2 and the UR5e is through ROS. The ROS ecosystem is the most widely used middleware in the robotics community and is based on the publisher/subscriber communication model. Thus, ROS is ideally suited to the system architecture [33]. Figure 4 shows a user operating the UR5e cobot through Microsoft Hololens 2. The user is indicating a point on a trajectory in space with his index finger.

Software using the tools offered by ROS has been developed with the purpose of keeping different nodes running in parallel to perform tasks. Likewise, this mechanism maintains the possibility of adding devices that synchronize with the data flow, thus maintaining the scalability and flexibility of the system. The instructions for the cobot are transmitted through the Real-Time Data Exchange (RTDE) interface. This interface is used to control and monitor the robotic arm.

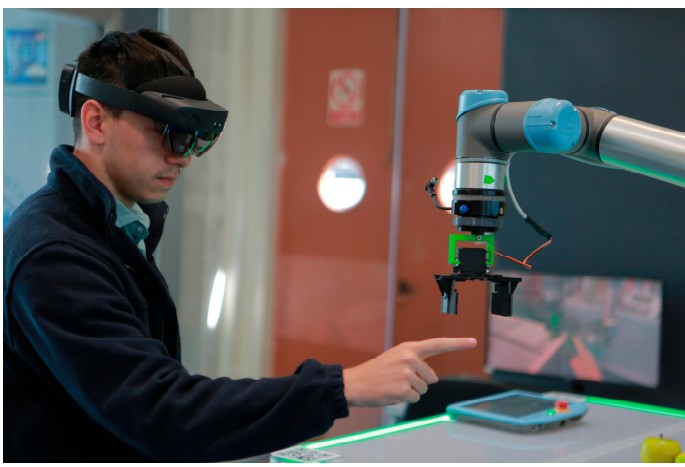

**Figure 4.** User interacting with the UR5e through Microsoft HoloLens 2.

### 4.1. Data Transmission

The communication infrastructure is established at system start-up, connecting the devices through a shared network. When the user starts the application on the HL2, it interacts with digital objects to plan trajectories while receiving real-time information about the state of the cobot. The specified coordinates are sent as messages to the intermediary. These messages are collected and used to generate the program that is sent to the cobot.

Communication and information flow in the system occurs through a specific set of nodes. Each of these nodes plays a crucial role in connecting, transmitting and receiving data between the different components of the system. These include:

- /ur_data: This node publishes UR5e information through RTDE.
- /hl2_client: Software that runs during the execution of the application in HL2. This node publishes messages from the specified coordinates by the user and subscribes to topics where it publishes/ur_data to show the user information about the status of the cobot.
- /rosbridge_websocket: A server that allows the connection of devices via websocket with the IP address of the host and port 9090. This node allows the connection of the/hl2_client node to ROS.
- /hl2ur5e: This node subscribes to the messages published by the/hl2_client node. It uses the published information to generate and send to the cobot the specified instructions by the user via RTDE.

The behaviour of this system is roughly represented in Figure 5.

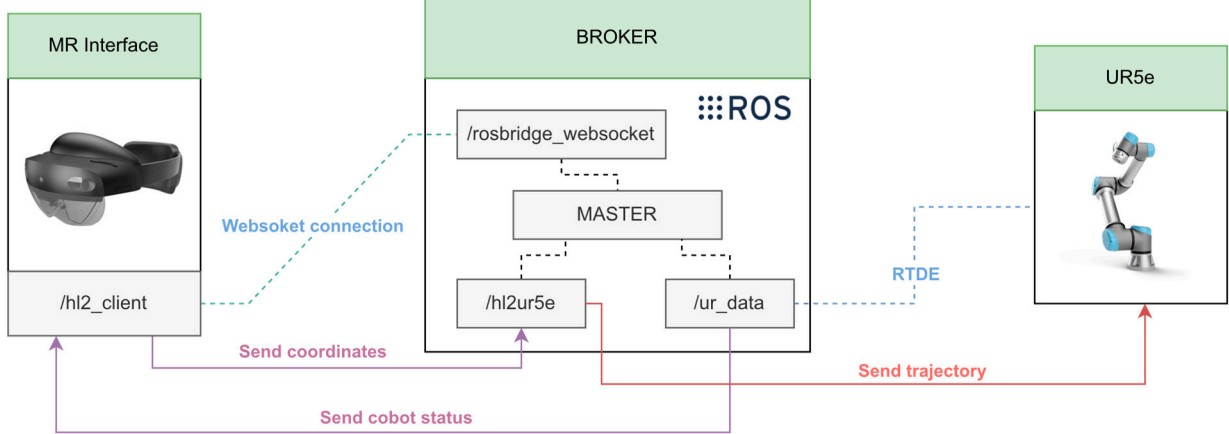

**Figure 5.** Conceptual architecture of system communication. The blue flashing lines indicate the connections between devices and the arrows indicate the direction of data flow.

### 4.2. Navigation through the MR Application

During the execution of the MR application, the user has visual access to digital elements projected on the lens of the HL2. These elements are composed of interactive dialogs with buttons, an interface with control buttons and a menu with various activity modes. Through voice and gesture recognition, the user can interact with these digital objects, either by activating predefined functions in the application scripts (such as showing or hiding digital elements) or by posting messages in the ROS topics. In addition, the user has the ability to move and rotate these objects using gestures.

HL2's gesture recognition relies on a computer vision algorithm and data acquired through depth sensors [34]. The algorithm effectively captures the skeletal structure of the user's hands, provided they are visible to the device's cameras. Leveraging the MRTK hand tracking profile, specific actions can be assigned for execution upon the detection of particular gestures [35].

Another relevant aspect is the recognition of voice commands. Keywords are defined to process them and execute the corresponding functions, making it easier to navigate within the application and perform actions in a versatile manner. The voice input functionality is driven by the identical engine supporting speech in all Universal Windows Platforms (UWP). The MRTK voice command profile allows you to configure the keywords to be recognized. This solution offers a wide range of applications, from point-to-point path planning to pick and place operations. Both types of manoeuvre can be automated, which significantly increases the complexity of the manoeuvres that are executable by the cobot.

### 4.3. Study Case

This section details the implementation of an object pick and place activity using the proposed solution. When the system is started, the connections are checked and the workspace is defined. This definition is carried out by detecting three QR codes placed on the cobot's worktable. These codes contain crucial information that the mixed reality application uses to build the matrix $^{W}T_{H}$ accurately and position the UR5e digital workspace in the real environment. When the QR codes are scanned, the application projects a digital 3D model in the form of a trihedron on each of them, allowing the user to confirm the correct execution of this process. The 3D coordinates defining the centres of these trihedrons coincide with the points **p1**, **p2** and **p3** previously specified in Section 3.2. Each QR code incorporates a unique message intended for precise construction of the $^{W}T_{H}$. This approach allows the user to scan the QR codes in any sequence, without affecting the correct derivation of the matrix. Figure 6 shows the process of reading a QR code where it can be seen how the user observes in the HMD lens digital elements that help him or her during the process.

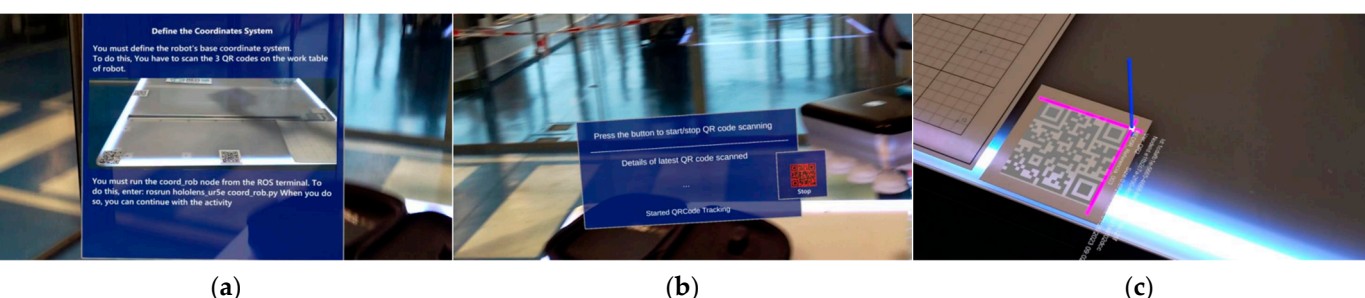

    (**a**)                           (**b**)                           (**c**)

**Figure 6.** QR code detection process. Screenshots of the application during its execution: (**a**) Instructions panel; (**b**) Start of detection; (**c**) QR code detected and recognized.

After this initial phase, the application projects a graphical representation of the UR5e operation space; this is located in the centre of its base. The visualization of this space, by default, is shown in green, as illustrated in Figure 7.

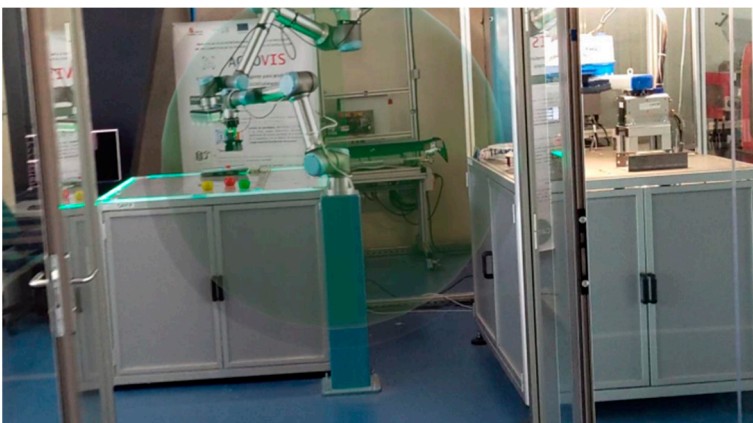

**Figure 7.** Visualization of the cobot workspace. Screenshots of the application during its execution.

During this stage, the user chooses the type of manoeuvre he or she wishes to perform. When the "Pick and place" option is selected, a menu appears with the necessary instructions to plan the trajectory correctly. Within this menu, the user is instructed to point with his or her index finger to the position of the object to be picked up by the cobot. Then, the user pronounces the voice command "Pick" to store the specified coordinate.

When pointing out the pick-up position of the object, this coordinate is analysed to verify its location within the workspace. In addition, the feasibility of its inverse kinematics to reach this position is evaluated. In the case of UR5e, the working volume is defined as a cylinder of 0.151 m diameter and 1.621 m height inside a sphere of 1.7 m diameter, following the example described in Section 3.2. Therefore, Equation (7) is used to verify that the coordinate is within the UR5e workspace. Thus, the thresholds dc and ds correspond to the values 0.151 and 1.7, respectively. The feasibility of inverse kinematics is performed through the functions provided by the RTDE interface.

If any of the above conditions is not met, the digital element representing the workspace changes its colour to red. A message is also issued to notify the user of this situation. This ensures the accuracy of the coordinates sent to the UR5e and allows the user, once the execution of the manoeuvre has started, to position himself or herself outside of this workspace for safety reasons.

If the conditions are met, the specified coordinate is stored in the memory and published in one of the ROS topics subscribed by the/hl2_client node. The/hl2ur5e node receives the coordinates and transforms them to the cobot reference, generating in parallel the instructions in RTDE format. Once the program is complete, it is sent to the cobot and it starts to perform the programmed trajectory to pick up the object.

During the movement of the UR5e, the/ur_data node monitors the speed variation of its joints and informs the application. This information is used to change the colour of LED lights placed on the cobot's workbench. These lights are used as beacons to indicate to the user when the robotic arm is in motion, thus improving safety during the activity.

Once the cobot stops its movement, the user can indicate the destination position in a manner similar to the pick-up position. Then, the application projects a panel that offers the user the option of performing a different manoeuvre or repeating the last program in a loop. This process is depicted in Figure 8. In this figure, it can be seen how the user indicates with an index finger the pick-up and destination positions of the object. The user enters these coordinates into the system by means of two voice commands: "Pick" and "Place". The application projects a yellow sphere on the saved locations to inform the user of the captured location. The/hl2ur5e node sends to the UR5e the program with the manoeuvres to pick and place the object in the specified positions. A video showing how the user performs this process can be found in the supplementary material section.

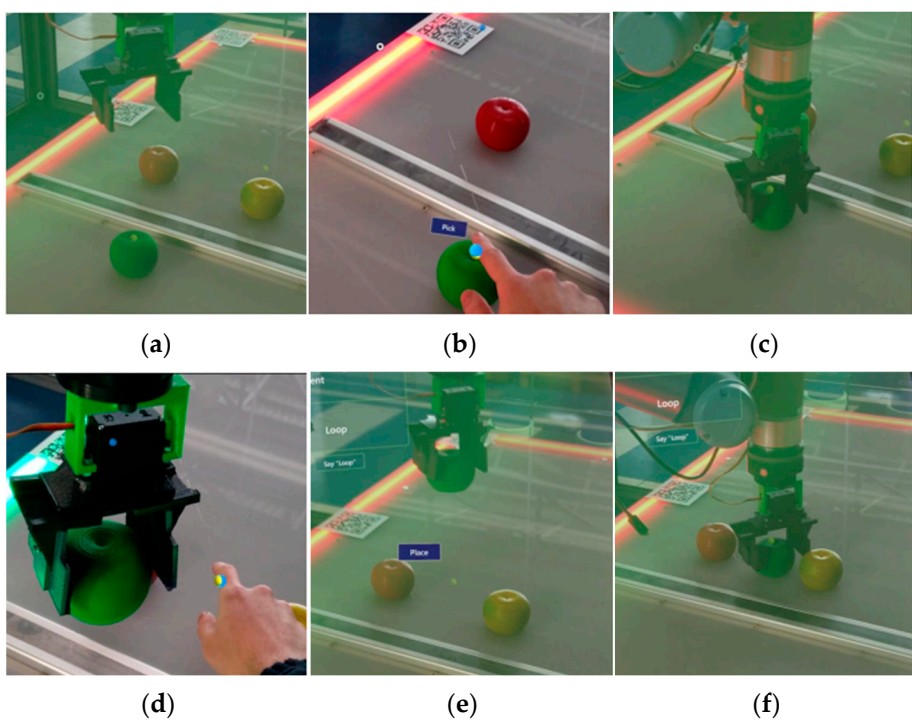

**Figure 8.** Experimental test of the pick and place mode. Captures of the application during its execution: (**a**) Cobot at rest; (**b**) Indication of the pick-up location; (**c**) Cobot picking up the object; (**d**) Indication of the target position; (**e**) Cobot moving the object to the target position; (**f**) Cobot placing the object.

*4.4. System Validation*

Following the objectives defined in Section 2, detailed tests have been carried out addressing various aspects of the system, and evaluating its performance according to the established criteria.

Preserving operator safety is a high priority requirement in environments where human–robot interaction is imperative. Although robotic arms provide a foundation of safety, the process requires additional tools to facilitate decision making [36]. Therefore, visualisation of the UR5e operating range is implemented in the HMD lens to ensure a better perception and understanding of the system's behaviour. Complementarily, a module of LED lights has been integrated, which are synchronized with the movements of the cobot, which are aimed at alerting the user about the operation status. Although the evaluation of this test has focused on a qualitative measurement, its objective is to reinforce the reliability of the system in the context of human interaction with the cobot.

The system's communication system operates using the TCP/IP protocol, ensuring highly reliable and virtually instantaneous data transmission. The command is transmitted virtually instantaneously from the moment it is issued until the robotic arm begins its movement. This ensures precise synchronization between the user's instruction and the action executed by the cobot.

In line with the papers [24,25], 10 volunteers' experience with the MR interface has been evaluated through a predefined activity. This aspect has been evaluated using the System Usability Scale (SUS). The SUS provides an overview of subjective usability evaluations by means of a survey. The survey comprises ten questions utilizing a five-point Likert-type scale that spans from "Strongly Disagree" to "Strongly Agree," with assigned values ranging from one to five, respectively. The statements are organized in a way in which odd-numbered items indicate a higher level of usability for the proposed system, while even-numbered statements imply the opposite. These questions cover various aspects such as the system's effectiveness, efficiency and user satisfaction. To obtain the final SUS

score, the total rating is multiplied by an appropriate factor to scale it within a range of between 0 and 100 [37].

The survey results detailed in Table 2 provide insight into the efficiency and improvement of the user experience with the application. Similarities can be found between the MR interfaces proposed in the papers mentioned above and the proposed solution. All three interfaces achieve a similar result in measured standard deviation. This may be due to the familiarity of the users with this type of technology. It can be said that the cause of this aspect has its origin in question 3. In this question, it can be observed that there is a great disparity in the mode of difficulty that the users noticed when carrying out the activity. However, it can be considered that the results obtained reflect the intuitive nature of the MR interfaces which, thanks to voice programming and user gestures, allow complex tasks to be carried out with the UR5e, surpassing in quality and efficiency the hand-guided methods.

**Table 2.** SUS scores measured.

| Question | Score (1–5) | Standard Deviation |
|---|---|---|
| I think that I would like to use this system frequently | 4 | 0.94 |
| I found the system unnecessarily complex | 2.20 | 1.03 |
| I thought the system was easy to use | 3.80 | 1.20 |
| I think that I would need the support of a technical person to be able to use this system | 2.50 | 0.71 |
| I found the various functions in this system were well integrated | 4.30 | 0.48 |
| I thought there was too much inconsistency in this system | 1.90 | 0.57 |
| I would imagine that most people would learn to use this system very quickly | 2.80 | 0.63 |
| I found the system very cumbersome to use | 1.70 | 0.67 |
| I felt very confident using the system | 4.70 | 0.48 |
| I think that I would like to use this system frequently | 4 | 0.82 |
| Mean (0–100) | 63.80 | 15.02 |

It has been proven that the system can reduce the programming time considerably. In different tests in which the programming time of a pick and place was timed, the manual guided programming method was compared with the proposed system. These tests showed that the proposed system can reduce the programming time by up to 80%. Clearly, this value is influenced by the robotics knowledge of the operator and his or her ability to build a script with the cobot's teach pendant.

It is also desirable to obtain the precision and accuracy of the system. The study by Soares et al. [38] has concluded that HL2 is suitable for applications in which the accuracy does not need to be less than 1 cm. However, this study has performed its own analysis working with the designed application.

The experiment focused on the application of the trajectory planning method for point capture. Seven volunteer participants were asked to place their index finger on five different points, which had been previously measured using the UR5e as a measuring instrument. Since the UR5e has a repeatability of 0.03 mm, these measurements made by the cobot were considered as the actual measurements of the experiment.

Each user-recorded coordinate was transformed to the collaborative robot's reference frame and compared to the actual measurement previously obtained from the cobot's teach pendant. This procedure was repeated 10 times for each designated location, generating a set of 350 measurements for each of the five volunteers. The data, presented in Table 3, reveal notable disparities between users and categories, suggesting possible variations in model performance depending on the user. These discrepancies hardly affect the test results. It is observed, for example, that with user U4 the system captures the coordinates of the index finger unsteadily, with a variation of up to 1 cm. In terms of accuracy, a remarkable similarity is recorded among all users, with user U1 performing the best, with a difference of 4.6 mm compared to user U5. It is noteworthy that users encountered increased difficulty in indicating the P2 and P3 positions compared to other positions. This heightened difficulty

may stem from factors such as poor visibility of the environment in this specific location; this is possibly caused by excessive light reflection or the user's posture.

**Table 3.** Precision and accuracy in the volunteer assays.

| | Precision (mm) | | | | | | Accuracy (mm) | | | | | |
|---|---|---|---|---|---|---|---|---|---|---|---|---|
| User | P1 | P2 | P3 | P4 | P5 | Partial Average | P1 | P2 | P3 | P4 | P5 | Partial Average |
| U1 | 2.13 | 11.15 | 0.77 | 3.11 | 1.47 | 3.73 | 12.90 | 11.87 | 10.29 | 9.68 | 6.60 | 10.27 |
| U2 | 2.15 | 11.39 | 1.00 | 0.79 | 1.06 | 3.28 | 12.17 | 12.37 | 7.25 | 9.08 | 9.08 | 9.99 |
| U3 | 1.48 | 2.36 | 1.81 | 4.93 | 1.29 | 2.37 | 7.25 | 10.45 | 10.46 | 11.61 | 11.37 | 10.23 |
| U4 | 1.98 | 8.88 | 33.79 | 2.18 | 4.25 | 10.22 | 7.15 | 14.64 | 25.73 | 8.80 | 10.25 | 13.31 |
| U5 | 1.29 | 4.25 | 7.41 | 0.98 | 2.40 | 3.27 | 12.14 | 20.23 | 26.29 | 8.61 | 7.15 | 14.88 |
| U6 | 1.79 | 8.51 | 3.56 | 2.12 | 1.71 | 3.54 | 8.37 | 13.4 | 17.68 | 8.42 | 7.98 | 11.17 |
| U7 | 2.23 | 7.54 | 10.14 | 7.11 | 1.24 | 5.65 | 12.12 | 13.68 | 18.1 | 14.42 | 5.65 | 12.79 |
| Global | | | | | | 4.58 | | | | | | 11.81 |

On the one hand, precision is a metric that strongly influences the performance of the application, as it is based on the HL2's ability to obtain similar measurements when environmental conditions do not change. On the other hand, accuracy is influenced by the user's skill; as each user's individual error is affected by his or her spatial perception when attempting to place the small blue sphere that tracks the position of the index finger at each location.

On further analysis, the precision of the system was determined to be 4.58 mm. This value was calculated by averaging the standard deviations of the points recorded by each user at all locations. Another interesting measure obtained is the accuracy which, with a value of 1.18 cm, corroborates the results obtained in Soares' study. This metric represents the mean of each of the absolute errors of each user. Therefore, it can be said that the performance of the system tends to improve as the user's experience with it increases.

*4.5. Discussion of Results*

The visualization of the cobot's reach area through the lens of the HL2 provides the operator with a better perception of the robotic arm's behaviour. This feature, coupled with the coordinate verification algorithm, promotes a safe working environment and minimizes errors. However, the possibility of operator interference in the working area of the cobot during the execution of a manoeuvre has not been evaluated. Although the UR5e stops its motion when colliding with objects or people, there is no guarantee that these will not be damaged upon impact. Monitoring the position of the operator's head and hands during manoeuvres has been considered; however, it is recognized that this approach may not be sufficient to prevent intrusions into the range area. To solve this problem, the scanning of other external elements such as 3D cameras or safety laser scanners is proposed. These devices could complement monitoring and decrease the risk of interference during cobot operations.

The human–robot interface greatly enhances the operator's ability to plan trajectories easily and accurately. Digital elements provide constant guidance during navigation, which facilitates task execution. Integrating attributes such as spatial audio or an assistance module to answer users' questions could significantly increase efficiency in planning tasks.

The results obtained during the precision and accuracy tests allow conclusions to be drawn about the possible applications of the system in the industrial environment. The metrics obtained during the test allow the system to be adapted to applications that are not restricted by this limitation. For example, the transfer of products from a conveyor belt to a container or the depalletizing of cartons. To compensate for this limitation, the MR application can be modified to allow the user to modify the saved point once it is captured.

Finally, it should be made clear that HL2 is not designed to be used for long periods of time. HMDs such as HL2 can cause eye strain, and neck and shoulder injuries if used

for long periods of time [39]. In addition, users immersed in MR may be less aware of their physical surroundings, increasing the risk of collisions or other accidents [40]. The inclusion of external elements, such as LED lights or the projection of digital information indicating the initiation of movements by the cobot, helps to mitigate the risks inherent to industrial activity related to the use of MR. For these reasons, the proposed solution is to plan manoeuvres quickly so that they are subsequently repeated in a loop and successively shut down the HMD. In this way, the solution can be adapted to assembly lines or repetitive execution tasks.

Based on the objectives defined in Section 2, Table 4 concisely presents each key contribution of the proposed solution. Comparing this table with Table 1, a significant improvement in aspects such as security, interface and programming time is evident with respect to the works reviewed in the state of the art. Accuracy and precision are similar to the study [20], with the particularity that the proposed system does not require continuous tracking of the marker position.

**Table 4.** Characteristics of the proposed solution according to the key aspects of MR for the industrial environment.

| Security | Communication | MR Interface | Planning Method | Range of Manoeuvres | Precision, Accuracy and Time |
|---|---|---|---|---|---|
| Visualization of the cobot's range Checking of coordinates and inverse kinematics RGB LED light module synchronized with the cobot | ROS WebSocket RTDE | Digital panels, dialogs and keypads with instructions Verification of detected QR code Superimposition of a blue sphere on the index finger | Capture of coordinates by index finger recognition Voice commands to save trajectory positions | Point trajectories Picking up and placing of objects | Precision of 4.57 mm Accuracy de 1.17 cm 80% reduction in programming time |

The study [23] describes a trajectory planning system that employs a workspace independent of the cobot. Consequently, it does not use a real reference to assess the measurements. Instead, it references previous work [30], in which accuracy better than 1 cm is achieved with the HL2. In addition, the range of manoeuvres allowed by this system is wider in comparison with the studies reviewed, which makes it possible to adapt it to different industrial manoeuvres.

The system presents a remarkable ease of scalability towards ROS2: the latest and most advanced version of ROS. The accelerated evolution of ROS2 is evident, especially due to its key features that are highly relevant in the industrial domain. The inclusion of the "ros1_bridge" package in ROS facilitates efficient bi-directional communication between ROS and ROS2. This makes it possible to take advantage of the distributed and real-time communication benefits provided by ROS2's Data Distribution Service (DDS) standard, thus improving interoperability and system responsiveness in dynamic industrial environments [41].

Future lines of work will focus on the optimization of the coordinates translation algorithm, so that other more complex manoeuvres can be planned. This will allow the system to be adapted to other industrial applications. We will also try to include new functionalities that allow it to operate with other cobot models, allowing the introduction of the positioning and work area parameters at the moment of workspace configuration. The system architecture is compartmentalized into modules that are each responsible for distinct tasks. The incorporation of emerging technologies such as generative artificial intelligence or cloud based data processing, has the potential to elevate the system to a higher operational standard.

## 5. Conclusions

The evolution of HMIs has been key to improve decision making and increase productivity [42]. In this context, the proposed solution contributes greatly to strengthen this development by enabling HRI through MR.

This research work has presented a tool for the manipulation of a UR5e robotic arm based on the use of an HMD as a process control and monitoring device. The added value of the solution is focused on the ability to plan trajectories in a fast and accurate way. The implementation of a human–robot interface that provides the user with digital information about the status of the cobot and assists the user in capturing the coordinates of the paths ensures the correct and safe execution of the planned manoeuvres. Results have shown that the precision of 4.58 mm and accuracy of 1.18 cm guarantee its application in most industrial applications.

The navigation experience through the human–robot interface is satisfactory. This allows users, regardless of their robotics knowledge, to plan each point of a trajectory by simply indicating it with their index finger. In addition, the application ensures that these points are reachable by the cobot, making use of a verification algorithm. A video details the operation of this activity, and shows a user marking a point in each corner of a box so that the UR5e performs a linear path passing through them. This video is available in the supplementary materials section.

The role of ROS is crucial in the system, as it is responsible for managing the communication between the devices in a reliable way. The computer programs that employ ROS transform the specified coordinates by the user into the format understood by the cobot. In turn, these programs analyse the state of the robotic arm to transmit the relevant information to the HMD.

The practical implications derived from this study on cobot scheduling are of considerable importance. The system's ability to reduce pick and place planning time by 80% has been validated. This achievement, therefore, brings substantial potential benefits to the industrial sector, including a notable increase in productivity and process efficiency. In addition, the application of MR can play a significant role in employee training, presenting an opportunity for companies to optimize both time and financial resources [43]. Additionally, the modularity of the proposed system allows the modification or incorporation of specific programs to perform different manoeuvres with different tools in the cobot. This process simply involves the corresponding coding to adapt these programs to the use of the information provided by the ROS topics.

In summary, the envisioned mixed reality system holds the potential for substantial advancements if implemented as a solution for industrial manoeuvre planning. Embracing emerging technologies such as MR is imperative in the industrial sector to outperform the competition and foster sustainability within the work ecosystem.

**Supplementary Materials:** The following supporting information can be downloaded at: https://www.mdpi.com/article/10.3390/electronics13030571/s1, Video S1: V1_Comparative_of_PickAndPlace.mp4; Video S2: V2_Trayectory_by_points.mp4.

**Author Contributions:** Methodology, R.C.-S., J.D.-D., J.G.-G.-B. and E.Z.; Software, R.C.-S., J.D.-D., J.G.-G.-B. and E.Z.; Validation, R.C.-S., J.D.-D., J.G.-G.-B. and E.Z.; Investigation, R.C.-S., J.D.-D., J.G.-G.-B. and E.Z.; Writing—original draft, R.C.-S., J.D.-D., J.G.-G.-B. and E.Z. All authors have read and agreed to the published version of the manuscript.

**Funding:** This work has been supported by the project "5R-Cervera Network in robotic technologies for smart manufacturing", contract number CER-20211007, under "Centros Tecnológicos de Excelencia Cervera" program funded by "The Centre for the Development of Industrial Technology" (CDTI); by the "Instituto para la Competitividad Empresarial de Castilla y León" (Project CCTT3/20/VA/0 0 03) through the line "2020 Proyectos I+D orientados a la excelencia y mejora competitiva de los CCTT", co-financed with FEDER funds; and ROSOGAR PID2021-123020OB-I00 funded by MCIN/AEI/10.13039/501100011033/FEDER, UE.

**Data Availability Statement:** Data are contained within the article.

**Conflicts of Interest:** The authors declare no conflict of interest.

## Abbreviations

| | |
|---|---|
| Abbreviations | |
| MR | Mixed Reality |
| HMD | Head Mounted Device |
| ROS | Robot Operating System |
| QR | Quick Response |
| HRI | Human–Robot Interaction |
| XR | Extended Reality |
| AR | Augmented Reality |
| VR | Virtual Reality |
| HMI | Human-Machine Interface |
| PbD | Programming by Demonstration |
| HL2 | Microsoft HoloLens 2 |
| 3D | Three-Dimensional |
| TCP | Tool Center Point |
| HTM | Homogeneous Transformation Matrix |
| HCS | Head Coordinate System |
| WCS | World Coordinate System |
| CCS | Cobot Coordinate System |
| MRTK | Mixed Reality Toolkit |
| RTDE | Real-Time Data Exchange |
| UWP | Universal Windows Platform |
| SUS | System Usability Scale |
| DDS | Data Distribution Service |

| | |
|---|---|
| List of Symbols | |
| $^C\mathbf{T}_H$ | Homogeneous Transformation Matrix of the HMD frame respect to the cobot frame |
| $\mathbf{p1}$, $\mathbf{p2}$, $\mathbf{p3}$ | Points in HMD frame |
| $\mathbf{p13}$ | Vector representing the distance and direction from point p3 to point p1 |
| $\mathbf{u}$, $\mathbf{v}$ y $\mathbf{w}$ | Unit vectors representing the directions of a 3D reference frame |
| $\mathbf{o}$ | Origin of a 3D reference frame |
| $^W\mathbf{T}_H$ | Homogeneous Transformation Matrix of the HMD frame respect to the world frame |
| $^C\mathbf{T}_W$ | Homogeneous Transformation Matrix of the world frame respect to the cobot frame |
| $^W\mathbf{T}_C$ | Homogeneous Transformation Matrix of the cobot frame respect to the world frame |
| $^W\mathbf{T}_C{}^{-1}$ | Inverse of a $^W\mathbf{T}_C$ |
| $^H\mathbf{p}$, $^C\mathbf{p}$ | Point obtained in the HMD and cobot frames |
| $dc$, $hc$ | Cylinder diameter and height |
| $ds$ | Sphere diameter |

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
