# Peer review of "Development of a Human–Robot Interface for Cobot Trajectory Planning Using Mixed Reality"

_electronics, doi:10.3390/electronics13030571_

Round 1
Reviewer 1 Report
Comments and Suggestions for Authors
Overall, I find the work to be promising and potentially significant for the field of collaborative robotics. The authors have presented a novel mixed-reality solution for programming complex tasks of collaborative robots (cobots) with a focus on efficiency, accessibility, and safety in human-robot interaction.
However, to enhance the quality and impact of the manuscript, I recommend addressing the following key points:
1. The manuscript lacks specific details regarding the implementation of the mixed-reality application and its integration into the mixed reality head-mounted device (HMD). Please provide more information on the technical aspects of the system, such as the algorithms employed for voice and gesture recognition, and the specifics of the digital panel interactions.
2. The authors claim positive feedback from users but do not provide concrete evaluation metrics or benchmarking against existing solutions. To strengthen the manuscript, include a detailed analysis of the system's performance, comparative studies with traditional programming methods, and discussions on the limitations observed during user evaluations.
3. Given the emphasis on safety in human-robot interaction, it is crucial to elaborate on the safety measures implemented in the proposed system. Discuss the potential risks associated with the use of mixed reality in an industrial setting and how these risks are mitigated to ensure the well-being of users.
4. The manuscript briefly mentions the use of QR codes to establish an accurate frame of reference. Elaborate on the calibration process for ensuring precise alignment between the mixed-reality application and the cobot. Discuss potential challenges and how they are addressed.
5. Given the potential impact on collaborative robotics, discuss the feasibility and challenges of industry adoption. Address how the proposed solution aligns with current industrial practices and whether any modifications would be required for seamless integration into existing workflows.
I believe that addressing these points will significantly improve the manuscript's clarity, rigor, and potential for acceptance. I look forward to reviewing the revised version of the manuscript.
Comments on the Quality of English LanguageMinor editing of English language required.
Reviewer 2 Report
Comments and Suggestions for Authors
The paper tackles the development of an interface for robot trajectory planning using mixed reality.
The system aims to optimize the interaction between the user and the machine and how the tasks are transferred to the robot.
The system is well described, and the videos attached to the paper are quite helpful.
The evaluation/validation is the main problem of the paper as it uses only six questions to evaluate the system and only 5users. A Technology acceptance model (TAM) may be a good addition.
The system should be compared with similar systems (because there are plenty) and see the differences. Some shared code on a GitHub can also be a plus.
Reviewer 3 Report
Comments and Suggestions for Authors
Dear Authors,
Your manuscript presents a significant contribution to the field of human-robot interaction and collaborative robotics. The integration of mixed reality (MR) in cobot trajectory planning is innovative and addresses a crucial need in the industry. Below are my comments and suggestions for improvement:
-
Abstract and Introduction Clarity: The abstract succinctly outlines the study's scope and significance. However, the introduction could benefit from a more detailed discussion on the challenges currently faced in cobot trajectory planning and how MR addresses these issues specifically. This will provide a clearer context for the study's objectives.
-
Literature Review: The review is comprehensive, but it would be beneficial to include more recent studies to ensure the manuscript reflects the latest advancements in this rapidly evolving field.
-
Methodology and System Analysis: The explanation of the system's architecture and the MR interface is well-articulated. However, a more detailed discussion on the selection of the MR equipment (e.g., why Microsoft Hololens 2 was chosen) and its specific advantages for this application would be insightful.
-
Experimental Setup and Results: The experiments conducted are robust and the results are promising. However, the manuscript would benefit from a deeper analysis of the results, particularly in comparing the performance of your system with existing methods. Additionally, including more detailed information on the participants' backgrounds in the experiments could help in understanding the user experience better.
-
Figures and Tables: The figures and tables are informative but could be improved for clarity. For example, labels and legends in the figures could be more descriptive. Also, consider adding a flowchart or schematic diagram illustrating the system workflow for better visualization.
-
Discussion on Limitations and Future Work: The discussion section is well-drafted, highlighting the system's potential and its limitations. However, it would be beneficial to expand on the future scope, particularly in terms of scalability of the system to other industrial applications and potential integration with other emerging technologies.
-
Technical Accuracy and Terminology: Ensure consistency in technical terms and acronyms throughout the paper. A glossary or a table of abbreviations at the beginning or end of the manuscript could be helpful for readers.
-
Conclusion: The conclusion succinctly summarizes the study's contributions. Consider emphasizing more on the practical implications of your findings and their potential impact on the future of cobot programming and MR applications in industrial settings.
-
References: Ensure that all references are up-to-date and relevant. Additionally, consider checking for any recent publications that might add value to your manuscript.
-
Formatting and Style: Check the manuscript for any formatting issues and ensure it adheres to the journal's submission guidelines. Also, proofread for grammatical and typographical errors.
Overall, your manuscript is well-composed and presents valuable insights into the application of MR in cobot trajectory planning. Addressing these comments will strengthen the paper and make a significant contribution to the field.
Comments on the Quality of English LanguageMinor Issues: Check the manuscript for any formatting issues and ensure it adheres to the journal's submission guidelines. Also, proofread for grammatical and typographical errors.
Round 2
Reviewer 1 Report
Comments and Suggestions for Authors
In the revised version of the manuscript, all my comments were considered. I thank the authors for that and appreciate their effort.